# Picornavirus Evolution: Genomes Encoding Multiple 2A^NPGP^ Sequences—Biomedical and Biotechnological Utility

**DOI:** 10.3390/v16101587

**Published:** 2024-10-09

**Authors:** Garry A. Luke, Lauren S. Ross, Yi-Ting Lo, Hsing-Chieh Wu, Martin D. Ryan

**Affiliations:** 1School of Biology, University of St. Andrews, Biomolecular Sciences Research Complex, North Haugh, St. Andrews KY16 9ST, UK; gal@st-andrews.ac.uk (G.A.L.); lauren.ross16@gmail.com (L.S.R.); 2International College, National Pingtung University of Science and Technology, 1, Shuefu Rd., Neipu, Pingtung 91201, Taiwan; geminiyiting@gmail.com (Y.-T.L.); jessiewu123@gmail.com (H.-C.W.)

**Keywords:** picornaviruses, polyprotein 2A region, 2A^NPGP^ sequences, ribosome skipping, translation, biotechnology

## Abstract

Alignment of picornavirus proteinase/polymerase sequences reveals this family evolved into five ‘supergroups’. Interestingly, the nature of the 2A region of the picornavirus polyprotein is highly correlated with this phylogeny. Viruses within supergroup 4, the *Paavivirinae*, have complex 2A regions with many viruses encoding multiple 2A^NPGP^ sequences. In vitro transcription/translation analyses of a synthetic polyprotein comprising green fluorescent protein (GFP) linked to β-glucuronidase (GUS) via individual 2A^NPGP^s showed two main phenotypes: highly active 2A^NPGP^ sequences—similar to foot-and-mouth disease virus 2A^NPGP^—and, surprisingly, a novel phenotype of some 2A^NPGP^ sequences which apparently terminate translation at the C-terminus of 2A^NPGP^ without detectable re-initiation of downstream sequences (GUS). Probing databases with the short sequences between 2A^NPGP^s did not reveal any potential ‘accessory’ functions. The novel, highly active, 2A-like sequences we identified substantially expand the toolbox for biomedical/biotechnological co-expression applications.

## 1. Introduction

Picornavirus genomes are single-stranded, positive sense RNA of some 7.5 to 8.5 kb in length. Their genomes have a common architecture: (i) a long 5′ untranslated region (UTR) containing an internal ribosome entry site (IRES), conferring an m^7^G cap-independent mode of the initiation of translation, (ii) a long open reading frame (ORF; ~2200aa ‘polyprotein’), and (iii) a short 3′ UTR preceding a poly(A) tail. In many genera, the N-terminal region of the polyprotein comprises a ‘leader’ sequence. The P1 domain of the polyprotein comprises four different proteins (1A-1D), 60 copies of each of which assemble to form the capsid. Proteins within the P2 (2A, 2B, 2C) and P3 (3A-3D) domains are replication proteins. Early studies showed the 3C protein to be a virus-encoded proteinase (3C^pro^), responsible for a ‘primary’ (co-translational) cleavage between the P2 and P3 regions, but also subsequent (‘secondary’) polyprotein processing, with 3C^pro^ being conserved amongst all picornaviruses (for review see [1]).

In the case of entero- and human rhinoviruses, a second virus-encoded proteinase, 2A^pro^ (~17 kDa), was identified and shown to be responsible for a single primary cleavage between the capsid proteins (P1) and P2 domains of the polyprotein—and, shown latterly, also to cleave key cellular proteins, thereby enhancing virus replication [2,3,4,5]. Simple inspection of the relatively modest number of picornavirus genome sequences available at that time—such as entero- and rhinoviruses, the foot-and-mouth disease virus (FMDV; aphthovirus), the encephalomyocarditis virus (EMCV; cardiovirus), and the hepatitis A virus (HAV; hepatovirus)—showed their 2A regions to be quite different from one another; indeed, the 2A protein of FMDV was only 18aa long. Research into polyprotein processing mechanisms of these other non-enterovirus genera showed that the capsid proteins domain was separated from the replication protein domains by either (i) the 3C proteinase cleaving the polyprotein in this region [6,7], or (ii) a proposed ‘ribosome skipping’ mechanism mediated by the 2A oligopeptide sequence proposed to interact with the ribosome exit tunnel, bringing about a discontinuity in the polypeptide backbone at the C-terminus of 2A—not by a proteolytic ‘cleavage’, but by ‘skipping’ the synthesis of a specific peptide backbone bond [7,8,9]. A completely conserved motif at the C-terminus of these 2A sequences (-NPG^↓^P-; position of discontinuity of the peptide backbone shown with arrow) gives rise to this type of picornavirus 2A being referred to in the literature as 2A^NPGP^. It should be noted, however, that 2A^NPGP^ (‘2A-like’) sequences are also found in a wide range of non-picornavirus virus families [10,11,12], plus non-LTR retrotransposons [13,14,15] and some cellular sequences (NOD-like receptor proteins—NLRs) [16,17]. Our previous work has shown that the ‘ribosome skipping’ activity resides within a ~25aa tract (the 2A C-terminal delimiter being -NPG^↓^P) [13,18]. Cardiovirus 2A proteins are ~143aa long; however, the C-terminal region of the cardiovirus-like 2As shares sequence similarity with the short aphthovirus-like 2As, both possessing the same proposed ribosome skipping function [19] outlined in Figure 1A.

It has been shown recently, however, that the cardioviruses EMCV and Theilers murine encephalitis virus (TMEV) 2A proteins are bifunctional in that they also stimulate a programmed ribosomal (−1) frameshift by binding an RNA secondary structure formed closely downstream—the first demonstration of protein-stimulated regulation of programmed ribosomal frameshifting. Here, a shift site occurs just 5′ of a stem-loop structure within the region encoding the 2B protein, producing a truncated 2B protein (2B*), effectively terminating translation prior to the translation of downstream replication proteins, which comprises some 50% of the entire polyprotein [20,21,22]. In the case of one genus, the dicipiviruses, however, the P1 region is encoded by an ORF separated from the second ORF (encoding the P2–P3 region) by a second, intergenic IRES that initiates translation of the downstream ORF [23].

Presently, the family *Picornaviridae* comprises almost 70 genera (Knowles www.picornaviridae.com; [24] accessed on 5 September 2024). Alignment and phylogenetic analyses of the nucleotide sequences corresponding to the uncleaved form ([3CD]) of 3C^pro^ and the 3D RNA-dependent RNA polymerase (3D^pol^) shows five major lineages, or ‘supergroups’ (SG), within the family: the *Caphthovirinae* (SG1), the *Kodimesavirinae* (SG2), the *Ensavirinae* (SG3), the *Paavivirinae* (SG4), and the *Heptrevirinae* (SG5) [25].

In the case of supergroups 2 (*Kodimesavirinae*—22 genera) and 5 (*Heptrevirinae*—7 genera), the separation between capsid and replication proteins is brought about by 3C^pro^ cleavage. In the case of supergroup 3 (*Ensavirinae*—8 genera), this separation is brought about by 2A^pro^, a proteinase unique to this supergroup. In supergroup 1 (*Caphthovirinae*—16 genera), this separation is brought about, in the vast majority cases, by a single copy of 2A^NPGP^. In the case of supergroup 4 (*Paavivirinae*—13 genera), different species encode either a single 2A^NPGP^ or multiple copies of 2A^NPGP^s.

In the case of the avihepatoviruses, typified by duck hepatitis virus type 1 (DHV-1), the 2A region comprises three proteins (Figure 1A): (i) 2A1 (20aa) is highly similar to the short aphthovirus 2A sequence—a highly efficient ribosome skipping sequence separating the capsid from replication proteins domains [26], (ii) 2A2 (161aa) has been shown to possess GTPase activity inducing apoptosis [27], and (iii) 2A3 (124aa) possesses similarity with parechovirus 2A, comprising an H-box/NC motif and related to the host-cell *H-rev* protein family [28], recently shown to promote cell proliferation [29]. In the case of Duck Picornavirus GL/12 (aalivirus A1) the complexity of the 2A region is increased, since it comprises six 2A proteins [30] (Figure 1A). Here, the 2A1 protein (19aa) is aphthovirus-like, 2A2 is 133aa, 2A3 is 150aa, and 2A4 is 131aa; all have C-terminal region (~25aa) similarity with aphthovirus-like 2A^NPGP^ sequences. Protein 2A5 has similarity with the avihepatovirus 2A2 protein, whilst 2A6 has similarity with the avihepatovirus 2A3 (parechovirus-like). Recently, the genome of the duck egg-reducing syndrome virus (DERSV) has been determined [31]. This virus encodes seven 2A proteins with six highly conserved 2A^NPGP^s.

Since a single iteration of 2A^NPGP^ is sufficient to separate encapsidation from replication proteins (e.g., SG1), the question arises, what function(s) do these ‘additional’ 2A^NPGP^- type 2As serve? The polypeptide tracts between successive 2A^NPGP^s are between ~50 and ~150aa long and could represent the acquisition of ‘accessory’ polypeptides. Multiple 2A^NPGP^s between the capsid and replication polyprotein domains could affect protein biogenesis, a key question being, what is the ribosome skipping activity of each of these sequences? Our previous work showed the activity of 2A/2A-like sequences resides within a ~25aa tract, including the N-terminal proline of the 2B downstream protein. In this study, we inserted a 2A/2A-like sequence (25aa, in-frame) between the green fluorescent protein (GFP: ~27 kDa—stop codon deleted) and β-glucuronidase (GUS: ~70 kDa), creating a single, long ORF [9,13].

It should be noted that certainly not all 2A-like sequences containing the conserved [D-V/I-E-X-NPGP] motif are active in ribosome skipping: the sequence immediately upstream, although not highly conserved, plays an essential role in ‘cleavage’. It is essential, therefore, to perform an assay to determine the activity of each sequence. Here, we used an in vitro transcription/translation system—the ‘skipping’ activity of each test sequence was determined by the incorporation of ^35^S-methionine into each translation product (Figure 1B). For viruses in supergroup 4, however, amino acid tracts between the 2A^NPGP^s are generally short; analysing the 2A region as a single tract could produce a significantly more complex mixture of translation products with either (i) poor resolution on SDS gels or (ii) low methionine content proving very difficult to detect, or (iii) producing translation products too small to be detected. We chose, therefore, to use our GFP/GUS system to determine the ‘skipping’ activity of these 2A^NPGP^s individually, rather than in their native, concatenated forms.

## 2. Materials and Methods

### 2.1. Bioinformatic Analyses

Picornavirus genome sequences were downloaded from the National Center for Biotechnology Information (NCBI) via the links provided on https://www.picornaviridae.com (accessed on 5 September 2024). The uncleaved form of the proteinase and polymerase ([3CD]) protein sequences were aligned using CLUSTALX [32]. The dendrogram file from CLUSTALX (.dnd) was opened using FIGTREE (http://tree.bio.ed.ac.uk/software/figtree/, accessed on 5 September 2024) to construct the phylogenetic tree. Polyprotein sequences were searched for occurrences of the ‘signature’ NPGP motif completely conserved amongst all 2A-like sequences to identify 2A^NPGP^s. Arbitrarily, 25aa tracts were chosen for further analyses (Table 1). Nucleotide and amino acid sequences of tracts between picornavirus 2A^NPGP^s were submitted to NCBI BlastN/BlastP (https://blast.ncbi.nlm.nih.gov/Blast.cgi, accessed on 5 September 2024) for database similarity searches.

### 2.2. Cloning of 2A^NPGP^ Sequences into pSTA1

All plasmids were constructed using standard methods and confirmed by automated nucleotide sequencing (Eurofins Genomics, Ebersberg, Germany). Restriction enzymes were purchased from Promega (Southampton, UK) and New England Biolabs (Hitchin, UK), whilst oligonucleotides were obtained from Eurofins Genomics (Ebersberg, Germany). PCR products encoding each 2A-like sequence were amplified from our pcDNA™ 3.1 mammalian expression vector, encoding ([pGFPF2AGUS]; pSTA1) [9] using the T7 ‘forward’ primer (5′-TAATACGACTCACTATAGGG-3′) and ‘reverse’ oligonucleotide primers listed in Table 2, such that a panel of GFP/2A-like PCR products was generated. Each PCR product was restricted with BamHI and ApaI, gel purified, and cloned into pSTA1, similarly restricted. Sequence identities were confirmed by automated DNA sequencing using an oligonucleotide primer corresponding to that encoding a C-terminal region of GFP (5′-CTGTCCACACAATCTGCCC-3′).

### 2.3. In Vitro Transcription/Translation

Plasmid DNA was linearised with *PstI* and purified using the Wizard SV system (Promega, Southampton, UK). Purified, linearised DNA (200 ng) was used to program a Wheat Germ Extract coupled transcription/translation system (Promega) supplemented with L-[^35^S]-methionine (EasyTag™ Perkin Elmer Beaconsfield, UK; 1 μL = 10 μCi) and the amino acid mixture (minus Met). The final reaction volume was adjusted to 25 μL using nuclease-free water. Reactions were incubated at 30 °C for 90 min before the addition of 2x SDS-PAGE loading buffer (Jena Bioscience, Jena, Germany). Translation products were analysed by 4–12% gradient SDS-PAGE (NuPAGE, Invitrogen, Paisley, UK) and the distribution of the radiolabel visualised by autoradiography.

## 3. Results

### 3.1. Bioinformatic Analyses

NOTE: Alignments and virus polyprotein sequences showing the positions of 2A^NPGP^s discussed below are shown in the Appendix A. Our alignment of [3CD] amino acid sequences produced the same phylogenetic relationships as the nucleotide sequence alignments reported by Zell and co-workers [25]. Our phylogenetic analyses show that picornaviruses encoding 2A^NPGP^s fall into two distinct supergroups—the *Caphthovirinae* (SG1) and the *Paavivirinae* (SG4), with viruses within the latter encoding either a single or multiple 2A^NPGP^s; indeed, sometimes this was the case for different species within the same genus (Table 1). Interestingly, there was a high correlation between each supergroup and the nature of the 2A region—the most plastic amongst picornavirus polyproteins (Figure 2). In the case of supergroup 1 (*Caphthovirinae*—16 genera), viruses encoded a single copy of 2A^NPGP^. The first exception here is mosavirus B1 which encoded two 2A^NPGP^s. In this virus, the first copy of 2A^NPGP^ appears to occur within the P1 region, whilst the second copy aligns with the single copy of mosavirus A 2A^NPGP^, which lies between the P1 and P2 polyprotein domains. The second exception is mupivirus, both A1 and A2, neither of which encode a 2A^NPGP^.

In the case of supergroup 4 (*Paavivirinae*—13 genera), variability was observed between different species within a genus. Here, a genus may comprise viruses which encode either (i) no 2A^NPGP^ (orivirus A1 and A2: alignment with avisiviruses shows a relative deletion in this region; see Appendix A), (ii) a single 2A^NPGP^ (aquamavirus, avihepatovirus, crohivirus, pasivirus, shanbavirus), (iii) multiple copies of 2A^NPGP^s—aalivirus, limnipivirus, Wuhan carp picornavirus (WCP—unassigned, but clusters within SG4) (NOTE: DERSV is most closely related to aaliviruses and clusters within the *Paavivirinae* (Figure 2)), (iv) genera comprising viruses encoding either a single or multiple 2A^NPGP^s (avisivirus, kunsagivirus, potamipivirus), or (v) genera comprising viruses encoding either no or multiple 2A^NPGP^s (grusopivirus, parechovirus; see Appendix A for alignments).

Wenzhou picorna-like virus 48 (WPLV-48; NC_032820) encoded three 2A^NPGP^ sequences but is unique in that 2A1 appears to lie between the P1 capsid and P2 replication protein domains, whilst 2A2 and 2A3 occur in the C-terminal region of the polyprotein—downstream of the WPL-48 sequences that align with the C-terminal 3CD region of all other picornaviruses (see Appendix A). In our alignment/phylogenetic analyses, this region was deleted. Although unassigned, WPLV-48 clustered within SG5 (Figure 2), the *Heptrevirinae,* which do not encode any 2A^NPGP^s (Figure 2). Interestingly, our 3CD alignment shows Wenzhou picorna-like virus 47 (WPLV-47; NC_033150) also clusters within SG5, although it does not encode a 2A-like sequence. Like many other viruses in this supergroup, WPLV-47 diverged at an early stage in the evolution of this supergroup.

For viruses with multiple 2A^NPGP^s, peptide sequences lying between each 2A^NPGP^ were used to probe the sequence database; other than similarities with related viruses, no significant matches were detected.

### 3.2. ‘Cleavage’ Activities of 2A-like Sequences

For all cases, in vitro translation profiles are shown in Figure 3, with estimated ‘cleavage’ activities shown in Table 3.

Alignments showing relative insertions/deletions in polyproteins within a genus, plus highlights of the position of 2A^NPGP^s, are provided in the Appendix A.

In the single instance of a virus clustering within SG1—mosavirus B1 encoding two 2A^NPGP^s—the activity of both sequences was directly comparable to FMDV 2A. The first mosavirus B1 2A^NPGP^ occurred within P1, whilst the second aligned with the single 2A^NPGP^ present within other mosavirus sequences and corresponded to the junction between the P1 and P2 domains of the polyprotein.

In the case of viruses encoding 2A^NPGP^s within SG4, however, a more complex pattern of activity was observed. Aalivirus A1/B1 2A^NPGP^s showed activities comparable to FMDV 2A, although aalivirus B1 sequences yielded slightly less of the GUS translation product. For both aalivirus A1 and B1 2A^NPGP^s, 2A1–2A4 (A1) and 2A1–2A5 (B1) produced similar translation profiles. Similarly, in the related grusopiviruses, different viruses encoded different numbers of 2A^NPGP^s. Grusopiviruses encoded a sequence (-TVIINPGP-) partly resembling 2A within the (predicted) P1 capsid protein domain of the polyprotein, although point mutations within the conserved C-terminal consensus of active 2A/2A-like sequences ([D(V/I)ExNPGP]) strongly suggest these are inactive. Alignment of grusopivirus polyproteins shows relative insertions/deletions within the 2A region are the cause of variable numbers of 2A^NPGP^s (see Appendix A). 2A1–2A3 of grusopivirus A1 and C viruses produced a similar translation profile to FMDV 2A, although GrV-C 2A2 showed a lower translation of the downstream protein, GUS. Again, relative insertions/deletions within the 2A region caused variable numbers of 2A^NPGP^s—notably virus YC-4, which only encoded one 2A^NPGP^ (see Appendix A). In the case of avihepatoviruses, related to aali- and grusopiviruses, all viruses encoded a single 2A^NPGP^. Avisivirus A1 and B1 viruses encoded two 2A^NPGP^s, both active but with no detectable ‘uncleaved’ [GFP2AGUS], and somewhat lower levels of GUS in comparison with FMDV 2A. Avisivirus A1 strain USA-IN1 and avisivirus C1, however, encoded a single 2A^NPGP^, whilst oriviruses did not encode a 2A^NPGP^—again due to relative insertions/deletions (see Appendix A). Wuhan carp picornavirus (WCP) encoded three 2A^NPGP^s: the translation profile produced by each sequence showed the [GFP2A] translation product but, surprisingly, no detectable ‘uncleaved’ [GFP2AGUS] or downstream GUS translation product. A similar translation profile was observed for the two potamipivirus B1 2A^NPGP^s, high [GFP2A] with low/no GUS translation, whilst potamipivirus A1 encoded a single 2A^NPGP^ due to relative insertions/deletions. The 2A^NPGP^s encoded by members of the limnipivirus genus showed a perhaps surprisingly wide range of translation profiles, whilst limnipivirus A1 2A1, type B1 2A2, type C1 2A1, and type D1 2A1–3 gave essentially the same translation profile as FMDV 2A; type A1 2A2 and type B1 2A1 produced the upstream translation product [GFP2A], but the downstream product, GUS, was not detected. Both crohivirus A and B encoded a single 2A^NPGP^, whilst in the parechoviruses, relative insertions/deletions in this region have produced a variable number of 2A^NPGP^s: type A has none, types B, C, D, and F have one, whilst types E and RtPV have two. Type E 2A1 only produced a full-length ([GFP2AGUS)] translation product, but type E 2A2 and RtPV 2A1/2A2 produced [GFP2A] as the major translation product with a low level of GUS. Shanba-, pasi-, aquama-, and kunsagiviruses encoded a single 2A^NPGP^, with the exception of kunsagivirus C, which encoded three 2A^NPGP^s—again arising from relative insertions/deletions in this genus. In the kunsagivirus polyprotein alignment, kunsagivirus C 2A3 aligned with the single 2A^NPGP^ within types A and B (see Appendix A).

Analyses of the ribosome skipping activities showed a mixed pattern (Table 3). Certain genera showed activities similar to the control FMDV 2A^NPGP^: aaliviruses, avisiviruses, grusopiviruses, mosavirus B1, and parechovirus 2A2. In a number of cases only [GFP2A] could be detected: kunsagivirus C1 2A1, limnipivirus A1 2A2, limnipivirus B1 2A1, potamivirus B1 2A2, WCP 2A1/2A2/2A3, and WP-LV48 2A1/2A2 and 2A3. In the case of parechovirus E, 2A1 was inactive, whilst parechovirus E 2A2, along with RtPV 2A1 and 2A2, was similar to FMDV 2A.

## 4. Discussion

### 4.1. 2A and 2A-like Sequences in SG1 and SG4 Viruses

Our phylogenetic analyses were based upon alignment of 3CD amino-acid sequences and showed a surprisingly high, but not complete, correlation with the nature of the 2A region of the polyprotein. It should be noted, however, that in the interpretation of these data, picornavirus mixed infections may show high levels of recombination (reviewed in [33]). Whilst SG1 comprises almost exclusively mammalian viruses, SG4 comprises ‘sub-lineages’ of avian (DERSV, aali-, avihepato-, avisi-, grusopi-, kunsagi-, ori-), fish (WCP, limnipi-, potamipi-), and mammalian (crohi-, parecho-, shanba-, pasi-, aquama-, kunsagi-) viruses. The picornavirus 2A region is highly plastic; viruses within SG4 may encode no 2A^NPGP^s (e.g., oriviruses), a single 2A^NPGP^ (e.g., aquama- and pasiviruses), or multiple 2A^NPGP^s. Indeed, viruses within the same genus may encode different numbers of 2A^NPGP^s (e.g., aali-, avisi-, kunsagi-, limnipi-, parecho-, and potamipiviruses). SG4 polyprotein alignments show that relative insertions/deletions within the 2A region determines the variable numbers of 2A^NPGP^s (see Appendix A).

In the case of SG1 viruses, the single 2A^NPGP^ sequence is responsible for a highly efficient ‘primary’, co-translational cleavage between capsid and replication proteins. Our in vitro transcription/translation analyses showed many 2A^NPGP^ sequences in SG4 to be as active as the FMDV 2A control: aalivirus A1/B1 2A1–2A4, grusopivirus A1/C 2A1–2A3, limnipivirus A1 2A1, B1 2A2, C1 2A1 and D1 2A1–3, and mosavirus B1 2A1, 2A2. We have shown that somewhat less efficient 2A^NPGP^ sequences can also be used for other virus genomes to acquire additional functional ‘modules’. For example, different RNA segments/proteins of certain rota- and totiviruses have acquired a [2A-like/dsRNA-binding protein] ‘module’ [10,34,35]. In the case of SG4 viruses with multiple iterations of 2A^NPGP^, could sequences between 2A^NPGP^s comprise ‘accessory’ protein(s)? Protein BLAST was used to probe databases using such sequences. These analyses only produced, however, significant matches against corresponding regions of closely related viruses, providing no indications of accessory functions. As outlined above, it has been shown that cardiovirus 2A proteins, ~143aa long, alongside the ribosome skipping activity, mediate programmed ribosomal frameshifting at a proximal site within protein 2B, such that the frameshifted ribosome quickly encounters a stop codon. Therefore, as the infectious cycle progresses and the level of the 2A protein increases, the expression of (downstream) replication proteins is progressively diminished. Remaining aminoacyl-tRNAs can be devoted, therefore, to the production of capsid proteins, thereby increasing the yield of particles.

### 4.2. Implications for Our Model of Ribosome Skipping

In our proposed model of 2A^NPGP^ ribosome skipping, the interaction between 2A^NPGP^ and the ribosome exit tunnel leads to the C-terminal portion of 2A^NPGP^ adopting a conformation within the P-site of the peptidyl transferase centre (PTC), such that tRNA^Pro^ in the A-site cannot form a peptide bond, since the imino group is sterically restrained, being within a ring structure—the phi angle is constrained. We proposed that the tRNA^Pro^ exits the A-site, allowing release factors 1 and 3 (eRF1/3) to enter this site to hydrolyse the bond between the nascent polypeptide and tRNA^Gly^ [7,8,9]. eRF1 exit from the A-site is promoted by eRF3 [36]. For the ‘pseudo-reinitiation’ necessary to allow the elongation cycle to continue, (i) the vacant A-site must be re-occupied by tRNA^Pro^ and (ii) the tRNA^Pro^ must be translocated from the A- to P-site by elongation factor 2 (eEF2), thereby allowing (iii) the next aminoacyl-tRNA to enter the A-site to recommence polypeptide elongation—in this case, the downstream replication proteins. Should any of these steps be inhibited, translation would be terminated at this point without ‘pseudo-reinitiation’—effectively, the phenotype we observed using our reporter system in the case of the kunsagivirus C1 2A1, limnipivirus A1 2A2, limnipivirus B1 2A1, potamivirus B1 2A2, WCP 2A1/2A2/2A3, and WP-LV48 2A1/2A2- and 2A3 2A-like sequences; we observed synthesis of [GFP2A], but no GUS.

Our model proposes that the C-terminal residues of 2A^NPGP^ adopt a conformation within the PTC such that a peptide bond with prolyl-tRNA (A-site) cannot be formed; it is possible that certain subset of 2A^NPGP^ sequences adopt a conformation that also precludes peptidyl-tRNA hydrolysis by eRF1, halting, rather than stalling, elongation. The translation of sequences upstream of 2A (GFP) without detectable translation of sequences downstream of 2A (GUS) has, perhaps, a parallel with ‘no-go decay’. It has been shown that stalled translation complexes can be ‘rescued’ by the activities of mammalian Pelota/Hbs1/ABCE1 [37]. Pelota is a molecular mimic of eRF1, although (i) lacks the -GGQ- motif of eRF1—the peptidyl-tRNA ester linkage is, therefore, not hydrolysed by Pelota, and (ii) binds into a vacant A-site in a stop codon independent manner; Hbs1 is a structural homologue of eRF3 (reviewed in [38,39,40]). It should be noted, however, that Pelota might act in the stop–go process in conjunction with other translational GTPases with which it interacts, e.g., GTPBP2 [41]. ABCE1 promotes dissociation of the ribosome subunits and nascent peptidyl-tRNA is released, with peptidyl-tRNA hydrolase (Pth) subsequently releasing tRNA from peptidyl-tRNA by cleaving the peptide/tRNA ester linkage. Consistent with our observations, this would produce [GFP2A] alone, although, following this analogy, the subsequent decay of cellular mRNA in the no-go decay pathway implies degradation of the virus RNA. The first caveat here is that our analyses of 2A-like sequences from a range of organisms were performed on a ‘synthetic’ reporter construct using an in vitro translation system; how representative is this of virus replication in these disparate hosts? In our previous studies, however, such in vitro analyses have shown that these data have been confirmed by their use in many biotechnological/biomedical applications, including heterologous expression in a wide range of organisms (see Section 4.3 below). The second caveat that arises from our choice of the analysis of 25aa tracts is that our previous work on FMDV 2A indicated that such a tract was highly active and commensurate with the cloning strategy for analysis of larger numbers of 2A-like sequences; it may be the case that other StopGo sequences require additional upstream sequences and/or host-specific tRNAs/translation factor abundances, etc., for maximal efficiencies.

The evidence that 2A^NPGP^ sequences function within the ribosome arises from two main sources. Firstly, construction of artificial polyproteins comprising two proteins, each of which bear N-terminal signal sequences. Here, sequences encoding the p40 and p35 subunits of IL12 were linked via FMDV 2A^NPGP^ to encode [p402Ap35] (p40 stop codon removed). Both subunits were secreted from the cell to form active IL12: the (formerly) N-terminal signal sequence of p35 was recognised by signal recognition particle (SRP) as a nascent N-terminal feature and p35 secreted from the cell, along with p40 [42]. This property has been observed for many other artificial polyprotein systems, e.g., heavy and light chains of monoclonal antibodies, T-cell receptor proteins, etc. [43]. Secondly, the 2A^NPGP^ ‘cleavage’ has been mapped to a 20–30aa tract that can be accommodated within the ribosome exit tunnel—one type amongst a family of ribosome arrest peptides (RAPs) [44]. Indeed, the C-terminal portion of a stalled nascent peptide in the ribosome exit tunnel has been shown to modulate the selectivity of the A-site [45].

Studies of 2A^NPGP^ activity using strains of *S*. *cerevisiae* with compromised levels of release factor activity showed in strains with eRF1 depleted and a greater proportion of ribosomes translated through the 2A coding sequence, whilst in strains with impaired eRF3 GTPase activity, many ribosomes failed to ‘pseudo-reinitiate’ and translate sequences downstream of 2A^NPGP^ [46]; these data led us to develop our model of 2A^NPGP^ ‘cleavage’ activity. In contrast, however, studies using reconstituted translation systems in vitro did not show any involvement of eRF1/3 [47]. In conclusion, this sub-set of 2A-like sequences that appear to terminate translation at 2A^NPGP^ poses a conundrum for both (i) our current model of 2A^NPGP^-mediated ribosome skipping and (ii) the effect of such sequences on the replication of kunsagi-, limnipi-, potami-, WCP, and WP-LV48 viruses.

### 4.3. Biotechnological and Biomedical Applications

2A/2A-like sequences have been used in a huge range of biotechnological and biomedical applications; these sequences have been shown to be active in all eukaryotic cell types (amoeba, yeast, fungi, algae, plant, animal) but not in prokaryotic cells (see [43]). In this paper, we report a series of 2A-like sequences that are directly comparable—in some cases superior—to those highly active sequences already in use: those encoded by aalivirus A1 2A1–2A4, aalivirus B1 2A1–2A4, avisivirus A1 2A2, avisivirus B1 2A1, grusopivirus A1 2A1–2A3, grusopivirus C 2A1/2A3, limnipivirus A1 2A1, limnipivirus B1 2A2, limnipivirus C1 2A1, limnipivirus D1 2A1/2A2, and mosavirus B1 2A1/2A2.

In the majority of cases, 2A^NPGP^s from different viruses have been used to co-express multiple different proteins (e.g., components of macromolecular structures, biochemical pathways, etc.) from a single ORF: for example, six different genes were co-expressed using five different 2A-like sequences to create autonomous bioluminescent human cells [48]. Alternatively, multiple iterations of the T2A peptide sequence with different codon usages were used to co-express nine different proteins comprising the carotenoid and violacein pathways in *Pichia pastoris* [49]. In both cases, however, constructs were designed to minimise the possibility of gene deletion via homologous recombination. In many cases this co-expression technology has replaced the need for time-consuming, costly, sequential transformations by the ability to link multiple genes into a single self-processing construct, thereby introducing a ‘trait’ by a single transformation: e.g., production of pluripotent stem cells, cancer gene therapies, CAR T-cell therapies, golden rice development, co-expression of glyphosate tolerance/BT toxins, introduction of novel biosynthetic pathways, etc. [42]. Here we describe a series of 2A-like sequences which substantially expand the toolbox for biotechnologists.

## Figures and Tables

**Figure 1 viruses-16-01587-f001:**
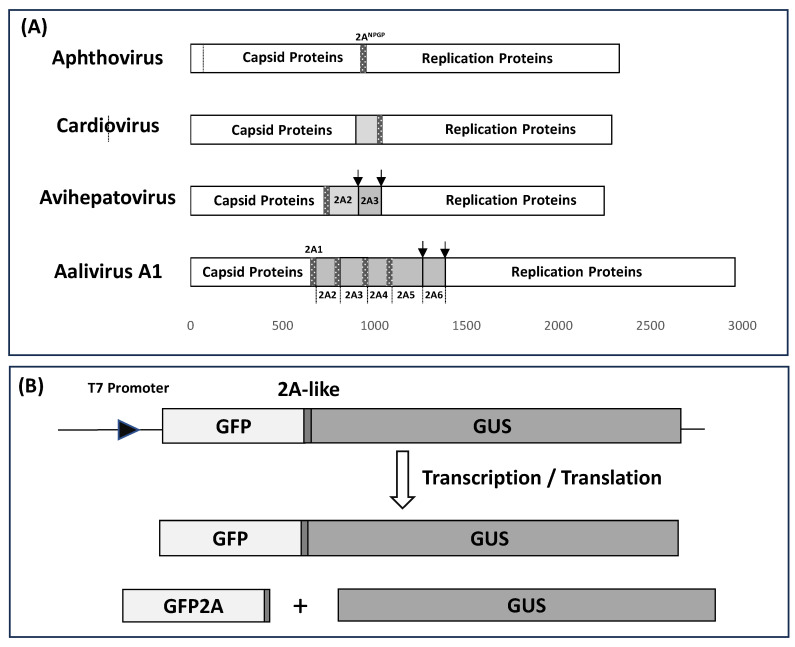
Schematic (drawn to scale) showing the positions of 2A^NPGP^ sequences (25aa) within the genomes of aphtho-, cardio-, avihepato-, and aalivirus polyproteins (Panel **A**). The [GFP2AGUS] artificial polyprotein (GFP stop codon removed) was used to test the various 2A^NPGP^ sequences. In the case of FMDV 2A^NPGP^, three translation products were observed: ‘uncleaved’ [GFP2AGUS] together with the ‘cleavage’ products [GFP2A] and GUS (Panel **B**).

**Figure 2 viruses-16-01587-f002:**
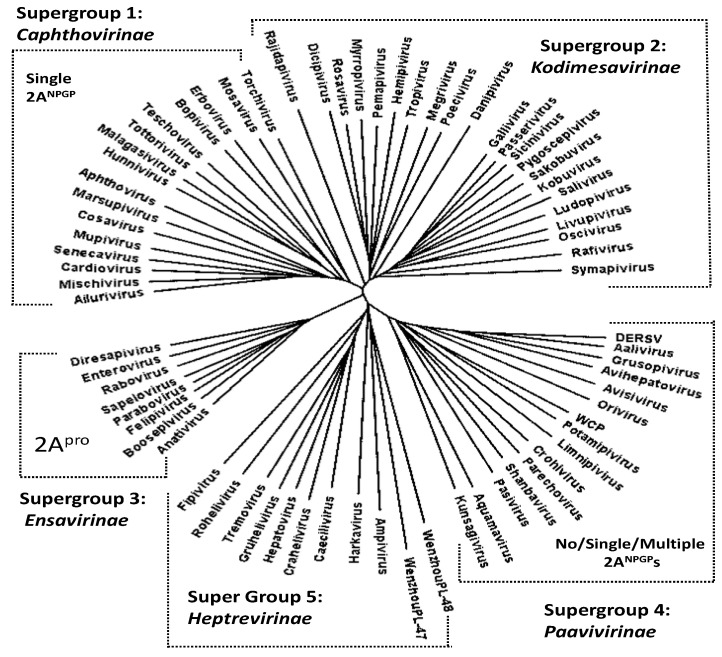
Dendrogram of aligned 3CD amino acid sequences rendered using FigTree. The supergroups indicated are consistent with those of Zell and co-workers [25]. Viruses encoding a 2A proteinase occurred only within supergroup 3, whilst 2A^NPGP^ sequences were found only within supergroups 1 and 4.

**Figure 3 viruses-16-01587-f003:**
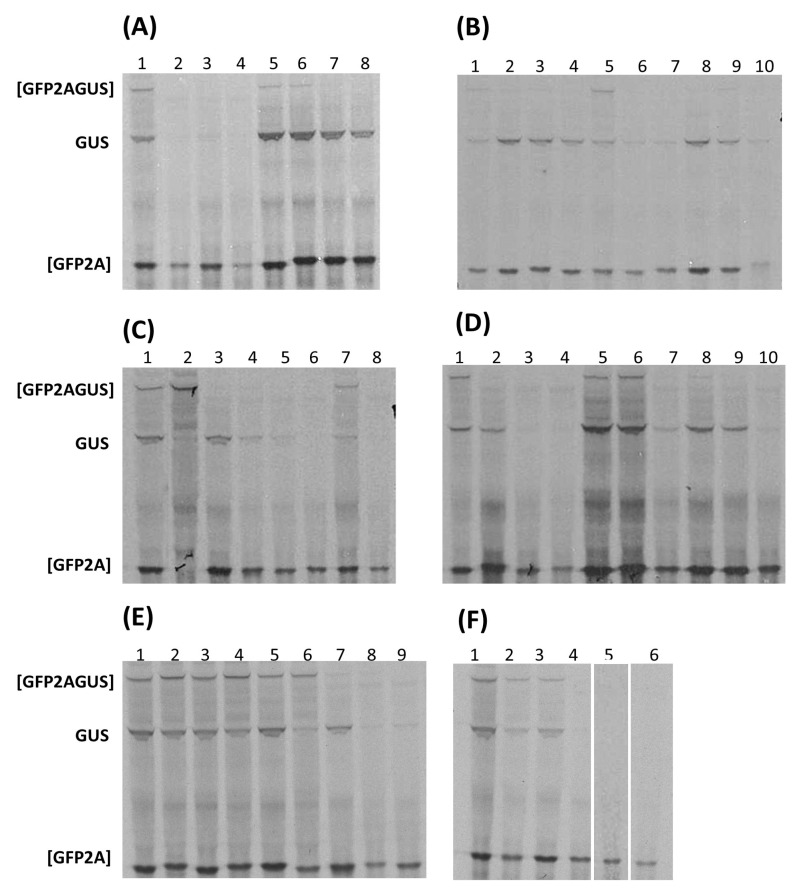
Translation in vitro. Coupled transcription/translation wheat germ extracts were programmed with the plasmid constructs indicated. Translation products were labelled with [^35^S] methionine, separated on 4–12% gradient SDS polyacrylamide gels, and detected by autoradiography. Gels are representatives of experiments performed in triplicate. Extended autoradiography did not reveal further visible translation products. Lane 1 panels (**A**–**F**) control construct FMDV 25 encoding [GFP-2A(25aa)-GUS]; positions of the uncleaved [GFP-2A-GUS] and cleavage products [GFP-2A] and GUS are indicated. (**A**) Lanes 2–4, WP-LV48: 2A1–2A3; lanes 5–8, aalivirus-A1: 2A1–2A4. (**B**) Lanes 2–6, aalivirus-B1: 2A1–2A5; lanes 7 and 8, avisivirus-A1: 2A1/2A2; lanes 9 and 10, avisivirus-B1: 2A1/2A2. (**C**) Lanes 2 and 3, parechovirus E: 2A1/2A2; lanes 4 and 5, RtPV: 2A1/2A2; lane 6, potamipivirus A1: 2A1; lanes 7 and 8, potamipivirus B1: 2A1/2A2. (**D**) Lanes 2 and 3, limnipivirus A1: 2A2 and 2A2; lanes 4 and 5, limnipivirus B1: 2A1 and 2A2; lanes 6 and 7, limnipivirus C1: 2A1 and 2A2; lanes 8–10, limnipivirus D1: 2A1, 2A2, and 2A3. (**E**) Lanes 2–4, grusopivirus-A1: 2A1, 2A2, and 2A3; lanes 5–7, grusopivirus-C: 2A1–2A3; lane 8, YC-4; lane 9, kungsagivirus-C1: 2A1. (**F**) Lanes 2 and 3, mosavirus B1: 2A1/2A2; lanes 4–6, WCP: 2A1–2A3.

**Table 1 viruses-16-01587-t001:** 2A^NPGP^ amino acid sequences and accession codes.

Genus	Species	2A^NPGP^	Amino Acid Sequence	Acc.#
Aphthovirus	FMDV O1K	2A	VAPVKQTLNFDLLKLAGDVESNPGP	GNNYF
Aalivirus (SG4)	AalV- A1	2A1	LLTSEGATNSSLLKLAGDVEENPGP	KJ000696
		2A2	FEMPYDDPEWDRLLQAGDIEQNPGP	
		2A3	PIPARPDPQWNNLQQAGDVEMNPGP	
		2A4	EHFNQTGGWVPDLTQCGDVESNPGP	
	AalV-B1	2A1	ATTLQVSEYLKDLTIDGDVESNPGP	MH453803
		2A2	LKVKKLEGDYVRDLTQEGVEPNPGP	
		2A3	SVRVTDAGWVRDLTVDGDVESNPGP	
		2A4	VFKCHDKCWVDDLTNCGDVEPNPGP	
		2A5	IFKCHEGCWVEDLTVDGDVESNPGP	
DERSV-AH204		2A1	TSTAQATSYVKDLTIDGDVESNPGP	UYL81882
		2A2	KTCREVEGSYVKDLTEEGIEPNPGP	
		2A3	LLKIGNAAWVRDLTEDGDVEENPGP	
		2A4	VYNCHESCWNRDLTIDGDVELNPGP	
		2A5	VFKCHEKCWQKDPTQDGDVEQNPGP	
		2A6	EFKCHEHCWVRDLTMDGDVEENPGP	
Avisivirus (SG4)	AsV-A1	2A1	EVGAYDEVDHRDILMGGDIEENPGP	KC465954
		2A2	EMGVFDETDHRDILLGGDIEENPGP	
	AsV-B1	2A1	PQFEKERSAHEDVLLGGDVESNPGP	KF979333
		2A2	SESVQYLEPQIDICVCGDVERNPGP	
Grusopivirus (SG4)	GrV-A1	2A1	FEKHVKPWRSQEDLSKEGIEPNPGP	KY312544
		2A2	ITDNRYKETDAKWLSRYGVEMNPGP	
		2A3	VTQDLYAATNQDQLSNQGIESNPGP	
	GrV-C	2A1	YFEERSPHPTQKELGQFGVETNPGP	MK443503
		2A2	ENNSNYSERDAKHLSRYGIEMNPGP	
		2A3	CVCTRWSPTMQSELGKYGIEKNPGP	
	YC-4	2A1	PERQYFSPKAKEELSKYGIEPNPGP	KY312543
Kunsagivirus (SG4)	Kuv-C1	2A1	IAAASAQGWQRDLTQDGDVESNPGP	KY670597
		2A2	LGIVISDSVWQRDLPREGVEENPGP	
		2A3	SYDPLAPSQWCRDLTCEGIEPNPGP	
Limnipivirus (SG4)	A1	2A1	CKEFVRESDNQELLKCGDVESNPGP	JX134222
		2A2	WDLSTGWFHFFRLLRSGDVEQNPGP	
	B1	2A1	MDVVDDYPFKRDLTRDGDVESNPGP	KF306267
		2A2	IDLVQAAYSRMRLLLSGDVEQNPGP	
	C1	2A1	KLLEQILAYKRDLTACGDVESNPGP	KF874490
		2A2	SRWIHARFARLRLLLSGDVEQNPGP	
	D1	2A1	EEEVDWGVGRMRLKMSGDVEENPGP	MG600094
		2A2	AVHLLVTWMRRRLTLSGDIESNPGP	
		2A3	DLRAVKSFIESQLMRAGDVERNPGP	
Mosavirus (SG1)	B1	2A1	ESRGTGNCDATTISQCGDVETNPGP	KY855435
		2A2	YVRRSANRTAADISQDGDVETNPGP	
Parechovirus (SG4)	E	2A1	WFDARTGFKTPLMNPCGDVEENPGP	KY645497
		2A2	QIEKRYGYRFWLLMLCGDVELNPGP	
	RtPV	2A1	MLDRRMGYRSRILCQCGDVEENPGP	MF352429
		2A2	WFNKRSGYRSRLLSQCGDVEENPGP	
Potamipivirus (SG4)	A1	2A1	IGEILTDKMWIRDLTQEGVEPNPGP	KC843627
	B1	2A1	LMEKTEEAGWLRDLTREGVEENPGP	MK189163
		2A2	FDDYHQEGGWIRDLTAEGVEPNPGP	
Unassigned (SG4)	WCP	2A1	MKEDEAGGWKEDLTEDGDVESNPGP	MG600066
		2A2	EQAIPETTWRRDLTQSGDVESNPGP	
		2A3	PGAIPASVWVHDLTTDGDVESNPGP	
Unassigned	WP-LV 48	2A1	GPSCYDRNNHCNILLSGDIEENPGP	NC_032820
		2A2	VFNASYLDCFISLLSCGDIESNPGP	
		2A3	PIQGLTQRFESTLLLGGDIEENPGP	

**Table 2 viruses-16-01587-t002:** Reverse oligonucleotide primers used to amplify GFP, thereby adding each 2A^NPGP^ 3′ extension; the ApaI restriction site used in cloning is indicated in bold typeface.

Genus	Species	2A^NPGP^	Reverse Primer Sequence (5′-3′)
Aalivirus	AalV- A1	2A1	GCGCGC**GGGCCC**TGGATTCTCTTCCACATCTCCAGCTAACTTTAACAGAC TTGAATTTGTGGCTCCCTCTGATGTGAGCAATCTAGACCCGGACTTGTAT
		2A2	GCGCGC**GGGCCC**AGGATTCTGTTCTATGTCTCCAGCCTGGAGCAGCCTGT CCCATTCTGGGTCATCATATGGCATTTCGAATCTAGACCCGGACTTGTAT
		2A3	GCGCGC**GGGCCC**TGGATTCATCTCAACATCACCAGCTTGCTGCAAATTAT TCCATTGTGGGTCAGGCCTGGCTGGAATTGGTCTAGACCCGGACTTGTAT
		2A4	GCGCGC**GGGCCC**GGGATTGGACTCTACATCACCACACTGCGTCAGATCGG GGACCCATCCCCCTGTCTGGTTGAAGTGCTCTCTAGA**C**CCGGACTTGTAT
	AalV-B1	2A1	GCGCGC**GGGCCC**AGGATTTGATTCAACATCTCCGTCAATGGTTAAATCTT TCAGATACTCAGACACTTGCAAAGTAGTTGCTCTAGACCCGGACTTGTAT
		2A2	GCGCGC**GGGCCC**AGGGTTAGGTTCCACACCCTCTTGAGTTAAATCTCTAA CATAATCTCCCTCAAGTTTCTTAACTTTCAATCTAGACCCGGACTTGTAT
		2A3	GCGCGC**GGGCCC**AGGGTTTGATTCCACATCTCCATCAACTGTGAGGTCTC TCACCCACCCAGCATCTGTTACTCTAACCGATCTAGACCCGGACTTGTAT
		2A4	GCGCGC**GGGCCC**TGGATTTGGCTCAACATCCCCACAATTCGTCAGGTCGT CAACCCAACATTTATCGTGGCACTTAAAAACTCTAGACCCGGACTTGTAT
		2A5	GCGCGC**GGGCCC**AGGGTTCGACTCCACATCACCATCAACAGTTAGATCCT CAACCCAACAGCCCTCATGACACTTAAAAATTCTAGACCCGGACTTGTAT
Avisivirus	AsV-A1	2A1	GCGCGC**GGGCCC**AGGGTTTTCTTCAATGTCACCCCCCATGAGAATGTCTC TGTGGTCCACTTCATCATAAGCTCCAACTTCTCTAGACCCGGACTTGTAT
		2A2	GCGCGC**GGGCCC**TGGATTCTCTTCAATGTCACCTCCAAGTAGTATGTCTC TGTGGTCAGTCTCATCAAAGACTCCCATCTCTCTAGACCCGGACTTGTAT
	AsV-B1	2A1	GCGCGC**GGGCCC**AGGGTTTGATTCTACATCTCCACCTAGCAGAACATCCT CATGGGCTGAGCGCTCCTTTTCAAACTGTGGTCTAGA**C**CCGGACTTGTAT
		2A2	GCGCGC**GGGCCC**GGGATTCCTCTCTACATCACCACAAACACAGATATCAA TCTGGGGCTCCAAATATTGAACAGACTCACTTCTAGACCCGGACTTGTAT
Grusopivirus	GrV-A1	2A1	**GGGCCC**AGGGTTTGGTTCAATTCCCTCCTTAGATAGATCTTCTTGTGATT TCCAAGGTTTCCCATGTTTTTCAAATCTAGACCCGGACTTGTATAGTTC
		2A2	**GGGCCC**TGGGTTCATTTCCACTCCATATCGGCTCAACCATTTAGCGTCGG TTTCCTTATAACGATTGTCCGTAATTCTAGACCCGGACTTGTATAGTTC
		2A3	**GGGCCC**AGGATTTGATTCAATGCCTTGATTTGATAACTGATCTTGATTAG TAGCAGCATAAAGATCCTGAGTGACTCTAGACCCGGACTTGTATAGTTC
	GrV-C	2A1	GCGCGC**GGGCCC**AGGATTAGTTTCTACTCCAAATTGCCCCAATTCCTTCT GAGTTGGATGTGGAGATCTTTCTTCAAAATATCTAGACCCGGACTTGTAT
		2A2	GCGCGC**GGGCCC**AGGATTCATCTCTATGCCATATCGTGATAAGTGTTTGG CATCTCTCTCAGAATAATTTGAGTTGTTCTCTCTAGACCCGGACTTGTAT
		2A3	GCGCGC**GGGCCC**TGGATTCTTCTCAATTCCATACTTACCTAATTCAGACT GCATGGTGGGACTCCACCTAGTGCAAACACATCTAGACCCGGACTTGTAT
	YC-4	2A1	GCGCGC**GGGCCC**AGGATTAGGCTCGATACCATATTTAGACAGTTCTTCCT TCGCCTTTGGAGAGAAATATTGACGTTCTGGTCTAGACCCGGACTTGTAT
Kunsagivirus	Kuv-C	2A1	GCGCGC**GGGCCC**AGGATTGCTCTCAACATCACCATCTTGAGTAAGGTCTC TTTGCCAGCCCTGTGCACTAGCCGCGGCAATTCTAGACCCGGACTTGTAT
		2A2	GCGCGC**GGGCCC**TGGATTTTCCTCAACACCTTCGCGGGGTAGATCCCGCT GCCACACAGAGTCGGAGATGACAATACCTAATCTAGACCCGGACTTGTAT
		2A3	GCGCGC**GGGCCC**TGGATTAGGCTCGATACCCTCACAAGTCAAATCCCTAC ACCACTGGCTGGGGGCCAGAGGGTCGTAGCTTCTAGACCCGGACTTGTAT
Limnipivirus	A1	2A1	GCGCGC**GGGCCC**TGGGTTAGACTCCACATCTCCACACTTGAGTAGCTCCT GGTTGTCTGATTCTCTTACAAATTCTTTGCATCTAGACCCGGACTTGTAT
		2A2	GCGCGC**GGGCCC**AGGGTTCTGTTCAACATCTCCGCTCCTCAACAACCGGA AAAAGTGAAACCATCCTGTTGAAAGGTCCCATCTAGACCCGGACTTGTAT
	B1	2A1	GCGCGC**GGGCCC**TGGGTTGCTCTCAACATCTCCATCACGTGTTAAGTCAC GTTTGAAAGGGTAATCATCAACGACATCCATTCTAGACCCGGACTTGTAT
		2A2	GCGCGC**GGGCCC**GGGATTTTGTTCAACGTCCCCCGATAACAACAACCTCA TGCGTGAGTAGGCAGCTTGCACCAAGTCGATTCTAGACCCGGACTTGTAT
	C1	2A1	GCGCGC**GGGCCC**AGGGTTGGACTCCACATCGCCACAAGCAGTCAAATCTC GCTTGTATGCCAGAATTTGTTCAAGCAGTTTTCTAGACCCGGACTTGTAT
		2A2	GCGCGC**GGGCCC**AGGGTTGGACTCCACATCGCCACAAGCAGTCAAATCTC GCTTGTATGCCAGAATTTGTTCAAGCAGTTTTCTAGACCCGGACTTGTAT
	D1	2A1	GCGCGC**GGGCCC**TGGGTTCTCCTCAACATCACCAGACATCTTCAGCCGCA TCCTGCCCACGCCCCAGTCGACTTCCTCCTCTCTAGACCCGGACTTGTAT
		2A2	GCGCGC**GGGCCC**TGGGTTGGATTCAATGTCTCCAGAAAGCGTCAATCGTC TGCGCATCCAAGTAACCAGTAAATGAACAGCTCTAGACCCGGACTTGTAT
		2A3	GCGCGC**GGGCCC**TGGGTTTCTCTCCACGTCACCAGCGCGCATCAATTGAC TTTCAATGAATGACTTCACTGCTCTTAAATCTCTAGACCCGGACTTGTAT
Mosavirus	B1	2A1	GCGCGC**GGGCCC**AGGATTGGTTTCAACATCCCCGCACTGACTGATAGTCG TCGCATCACAGTTTCCTGTGCCACGAGATTCTCTAGACCCGGACTTGTAT
		2A2	GCGCGC**GGGCCC**GGGGTTGGTCTCAACATCTCCATCCTGACTGATATCAG CGGCAGTACGGTTTGCGGACCGCCTGACGTATCTAGACCCGGACTTGTAT
Parechovirus	E	2A1	GCGCGC**GGGCCC**TGGGTTTTCTTCCACATCACCACAGGGGTTCATTAGGG GTGTTTTAAACCCCGTGCGTGCATCAAACCATCTAGACCCGGACTTGTAT
		2A2	GCGCGC**GGGCCC**AGGATTTAACTCAACATCTCCACAGAGCATTAGCAACC AGAAACGATAGCCATATCGCTTCTCTATCTGTCTAGACCCGGACTTGTAT
	RtPV	2A1	GCGCGC**GGGCCC**TGGATTTTCCTCGACATCTCCACATTGACAGAGGATTC TGCTCCGATAGCCCATTCTCCTGTCAAGCATTCTAGACCCGGACTTGTAT
		2A2	GCGCGC**GGGCCC**AGGATTTTCTTCTACATCACCACATTGAGACAACAATC TTGACCTGTATCCTGATCTTTTGTTGAACCATCTAGACCCGGACTTGTAT
Potamipivirus	A	2A1	GCGCGC**GGGCCC**TGGATTTGGTTCAACTCCTTCTTGTGTCAGATCTCTGA TCCACATCTTGTCCGTGAGTATTTCGCCAATTCTAGACCCGGACTTGTAT
	B	2A1	GCGCGC**GGGCCC**GGGGTTCTCCTCAACTCCCTCTCTTGTCAAATCTCTTA GCCATCCTGCTTCTTCTGTTTTCTCCATCAATCTAGACCCGGACTTGTAT
		2A2	GCGCGC**GGGCCC**CGGGTTGGGCTCCACACCCTCAGCAGTGAGGTCCCGTA TCCAACCACCTTCCTGGTGGTAATCATCAAATCTAGACCCGGACTTGTAT
Unassigned	WCP	2A1	CGCGCG**GGGCCC**AGGGTTACTCTCCACATCACCGTCCTCAGTGAGGTCTT CTTTCCACCCACCAGCTTCATCCTCCTTCATTCTAGACCCGGACTTGTAT
		2A2	GCGCGC**GGGCCC**TGGATTGGATTCCACATCACCAGATTGTGTGAGATCTC GACGCCATGTGGTTTCAGGAATTGCTTGCTCTCTAGACCCGGACTTGTAT
		2A3	GCGCGC**GGGCCC**AGGATTGGATTCAACATCACCATCTGTTGTGAGGTCAT GAACCCAGACACTTGCTGGTATGGCACCAGGTCTAGACCCGGACTTGTAT
Unassigned	WP-LV 48	2A1	GCGCGC**GGGCCC**TGGATTCTCTTCAATATCTCCTGAAAGTAAGATGTTGC AATGATTATTCCTGTCGTAGCAAGATGGACCTCTAGACCCGGACTTGTAT
		2A2	GCGCGC**GGGCCC**TGGATTTGACTCGATATCCCCACAAGATAATAAGCTGA TGAAACAATCTAAATAACTGGCATTAAAAACTCTAGACCCGGACTTGTAT
		2A3	GCGCGC**GGGCCC**TGGATTTTCTTCAATATCGCCCCCCAAAAGAAGAGTTG ACTCAAAACGTTGTGTAAGACCTTGTATTGGTCTAGACCCGGACTTGTAT

**Table 3 viruses-16-01587-t003:** Estimated ‘cleavage’ activities of 2A-like sequences.

Genus	Species	2A^NPGP^	[GFP2AGUS]	[GFP2A]	GUS
Aphthovirus	FMDV O1K	2A	+	++++	++
Aalivirus (SG4)	AalV-A1	2A1	+	+++++	+++
		2A2	+	+++++	+++
		2A3	−	+++++	+++
		2A4	−	+++++	+++
	AalV-B1	2A1	−	++++	++
		2A2	−	++++	++
		2A3	−	++++	++
		2A4	+	++++	++
		2A5	−	++++	+
Unassigned	DERSV	2A1–6	ND	ND	ND
Avisivirus (SG4)	AsV-A1	2A1	−	++++	+
		2A2	−	++++	++
	AsV-B1	2A1	−	++++	++
		2A2	−	++	+
Grusopivirus (SG4)	GrV-A1	2A1	+	++++	++
		2A2	+	++++	++
		2A3	+	++++	++
	GrV-C	2A1	+	++++	++
		2A2	+	++++	(+)
		2A3	(+)	++++	++
	YC-4	2A1	−	++++	(+)
Kunsagivirus (SG4)	Kuv-C1	2A1	−	++++	−
		2A2	ND	ND	ND
		2A3	ND	ND	ND
Limnipivirus (SG4)	A1	2A1	−	++++	++
		2A2	−	++++	−
	B1	2A1	−	+++	−
		2A2	++	+++	++++
	C1	2A1	++	++++	++++
		2A2	−	++++	+
	D1	2A1	−	++++	++
		2A2	−	++++	++
		2A3	−	++++	+
Mosavirus (SG1)	B1	2A1	+	++++	++
		2A2	+	++++	++
Parechovirus (SG4)	E	2A1	+	−	−
		2A2	−	++++	+
	RtPV	2A1	−	++++	+
		2A2	−	++++	+
Potamipivirus (SG4)	A1	2A1	−	++++	−
	B1	2A1	(+)	++++	+
		2A2	−	+++	−
Unassigned (SG4?)	WCP	2A1	−	+++	−
		2A2	−	+++	−
		2A3	−	+++	−
Unassigned (SG5?)	WP-LV 48	2A1	−	+++	−
		2A2	−	++++	−
		2A3	−	+++	−

## Data Availability

The picornavirus sequence datasets analyzed in this manuscript were accessed from The National Center for Biotechnology Information (NCBI—https://www.ncbi.nlm.nih.gov/ (accessed on 5 September 2024)). The polyprotein sequence alignments plus indications of the positions of 2A-like sequences are given within the Appendix A submitted in this article.

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
