# Peer review of "Picornavirus Evolution: Genomes Encoding Multiple 2ANPGP Sequences—Biomedical and Biotechnological Utility"

_viruses, 2024, doi:10.3390/v16101587_

Round 1
Reviewer 1 Report
Comments and Suggestions for Authors
Picornavirus 2A(NPGP) sequences occur primarily at the junction of structural and nonstructural protein-coding regions of the genome. They support a stop-go (termination-reinitiation) process in which the peptide bond to the terminal proline of the motif fails to form, releasing the upstream polypeptide. Subsequent reinitiation involves re-binding of prolyl-tRNA to the vacated ribosomal A site.
In the present report, Luke et al. established the phylogenetic distribution of 2A(NPGP) sequences in picornaviruses and then characterized the activity of a variety of these sequences, determining that this varies from readthrough without a ‘stop-go’ event to strong release of the upstream ORF with or without subsequent downstream reinitiation. This result is interesting because it shows that the “stop” and “go” stages of this process are separable. The experiments were conducted using small (75nt-long) fragments of the viral genomes inserted between GFP and GUS reporters, so it remains to be determined whether a similar failure to reinitiate after release of the upstream coding sequence occurs in the context of translation of the entire genome in infected cells. It is possible that the efficiency of readthrough and/or reinitiation is influenced by the sequence or structure of the coding sequence immediately downstream of the NPGP-encoding codons in some picornavirus genomes, but this remains to be determined.
The report is straightforward, but is compromised by the poor quality of some of the translation data, such that even the product of translation of the first ORF is barely visible e.g. in Figure 3A, lane 4; Figure 3B, lane 10, Figure 3C, lane 8; Figure 3E, lane 8; Figure 3F, lane 6), so that it is not possible to establish whether low-level readthrough or reinitiation had occurred. The extent of cleavage appears to have been underestimated in some instances because of the weak level of translation (e.g. Fig. 3B, lane 10). The Methods section does not describe how [GFP2A], Gus and [GFP2AGUS] levels were quantitated, nor how (or even whether) reproducibility was established. It is not evident whether the score for e.g. GUS in an experiment reflects its absolute prominence compared to GUS generated in other translation reactions or its prominence relative to GFP2A in the same experiment. Was a correction factor applied to take the very different levels of translation of the different mRNAs into account?
SPECIFIC COMMENTS
1. The Discussion of the possible role of Pelota/Hbs1 in recycling stalled ribosomal complexes should take into consideration observations reported in Ref. 37 indicated that they act only on ribosomal complexes stalled very near to the 3’-end of mRNA, possibly due to structural characteristics of Hbs1 (see PMID: 21623367, reviewed in PMID: 35147231). Is it possible that Pelota might act in the NPGP stop-go process in conjunction with other translational GTPases, for example GTPBP2 (PMID: 30108131) with which it interacts (PMID: 25061210)?
2. Figure 3C included Potamipivirus A1, but it is not included in the summary Table 3.
3. Lines 272, 277. Neither Table 1 nor the supplementary data show Crohivirus, Pasivirus or Aquamavirus sequences.
4. Lines 279-280. Kunsagivirus C 2A-3 is in fact the poorest match to the single 2A(NPGP) sequences from Kunsagivirus A and Kunsagivirus B (Kunsagi_A-2A-1 matches Kunsagi_C-2A-1 , and Kunsagi-B-2A-1 matches Kunsagi_C-2A-2 ):
# Percent Identity Matrix - created by Clustal2.1
1: Kunsagi_C-2A-2 100.00 48.00 54.17 41.67 41.67
2: Kunsagi_C-2A-3 48.00 100.00 45.83 37.50 33.33
3: Kunsagi_C-2A-1 54.17 45.83 100.00 44.00 36.00
4: Kunsagi_A-2A-1 41.67 37.50 44.00 100.00 48.00
5: Kunsagi-B-2A-1 41.67 33.33 36.00 48.00 100.00
5. Line 29. The IRES is part of the 5’UTR, but doesn’t comprise it, since the 5’UTR also contains elements devoted exclusively to replication.
6. Line 41. 2A proteinases cleave cellular proteins at specific sites, but don’t per se degrade them.
Comments on the Quality of English Language1. Grammatical errors
There are several examples of errors in subject-verb agreement (correct versions below):
Line 21: 2A-like sequences ...... expand
Line 252: insertions/deletions ....... are the cause
Lines 273-4: relative insertions/deletions in the region have produced ....
Line 301: alignments show that relative insertions/deletions with the 2A region determine ...
2. Typographical errors.
Lines 151-152 ....sequencing [done] by Eurofins
Line 262 due to [a] relative insertions/deletions
Line 267 due to [a] relative insertions/deletions
Reviewer 2 Report
Comments and Suggestions for Authors
This manuscript presents a fascinating and comprehensive survey of StopGo sequences across the Picornaviridae family, besides experimental analysis of ~50 different examples. Overall, the manuscript is well written, clearly explained, and provides interesting new data. As such, I only have minor comments.
o A weakness of the manuscript is that the GFP-GUS, GFP and GUS expression levels are simply estimated with a -/+/++/+++ etc point scale rather than quantified with densitometry. It would be nice to add more precise quantification if the authors are in a position to do so.
o As noted in the manuscript (lines 377-378), failure to re-start following peptide release at a StopGo site raises a conundrum for virus replication as the RdRp etc would not be produced. Presumably there must be some re-starting, at least in some conditions (cell type, host species, etc). It is interesting that StopGo is close to 100% efficient in FMDV and EMCV etc as, from a gene-expression-level point of view, a low RdRp:capsid ratio would seem to be advantageous. This suggests that the high efficiency may be more to do with preventing e.g. NoGo decay (mentioned by the authors on line 354-355) than controlling gene expression levels. While StopGo certainly fails at some level (e.g. 5-10%) during infection with these viruses, one wonders whether the cases with much lower efficiency measured in the reporter context are truly representative of the situation during virus infection. I appreciate that experimentally testing StopGo efficiencies in the context of virus infection is beyond the scope of the current manuscript. However, perhaps a little more discussion of the possibility that the results may not be representative of the StopGo efficiency during virus infection would be useful? E.g. even though 25 aa of upstream sequence was shown to be sufficient for the FMDV StopGo sequence, it may be that other StopGo sequences require additional upstream sequence and/or host-specific tRNAs/factor abundances etc for wildtype efficiencies.
o Virus vernacular names should normally use lower case (e.g. “foot-and-mouth disease virus” not “Foot-and-Mouth Disease Virus”) unless they include a proper noun.
o There are some places where minor typos/grammar issues could be tidied up (e.g. “referred to” not “referred-to” on line 56).
o Line 55: “shown” -> “shown with arrow”?
o Line 76: The EMCV/TMEV frameshift protein is actually called 2B* not 2B’. 2B’ is 2B with the N-terminus removed at a 3C-Pro proteolytic cleavage site just downstream of the frameshift site but encoded in the zero-frame.
o Lines 138-139: How were phylogenetic trees constructed?
o Line 248: Perhaps the meaning here would be clearer if the text “Grusopiviruses encode a sequence resembling 2A” was altered to “Grusopiviruses encode a sequence, TVIINPGP, partly resembling 2A”?
o Line 312: I suggest that as well as protein blast the authors also test these sequences in the HHpred webserver (https://toolkit.tuebingen.mpg.de/tools/hhpred) querying e.g. the Pfam and UniProt databases.
